# Enhanced Spin Thermopower in Phosphorene Nanoribbons via Edge-State Modifications

**DOI:** 10.3390/nano12142350

**Published:** 2022-07-09

**Authors:** Junheng Ou, Qingtian Zhang

**Affiliations:** 1School of Materials and Energy, Guangdong University of Technology, Guangzhou 510006, China; junheng_ou@163.com; 2Guangdong Provincial Key Laboratory of Information Photonics Technology, Guangdong University of Technology, Guangzhou 510006, China

**Keywords:** quantum transport, Seebeck coefficient, phosphorene nanoribbon, edge states

## Abstract

We investigated spin-dependent thermoelectric transport in zigzag phosphorene nanoribbons with a ferromagnetic stripe. We explored the possibility to enhance the spin thermopower via modifications of the edge states in zigzag ribbons. Two methods are proposed to modulate the edge transport: one is applying gate voltages on the edges; the other is including notches on the ribbon edges. The transport gap is enlarged by the edge-state modifications, which enhance the charge and spin Seebeck coefficients almost twofold. Our results suggest phosphorene to be a promising material for thermoelectric applications and open a possibility to design a tunable spin-thermoelectric device.

## 1. Introduction

Since the successful fabrication of graphene [1], two-dimensional materials have attracted considerable interest due to their potential applications in the next generation of nanodevices [2,3,4,5,6,7]. Among these materials, phosphorene, a monolayer of black phosphorus atoms arranged in a puckered honeycomb lattice [8,9], has been intensively studied [10,11,12]. Phosphorene has some advantages over other two-dimensional materials such as graphene and transitional metal dichalcogenides (TMDCs). Previous studies have shown that graphene has a high carrier mobility but zero band gap [13], and TMDCs have a band gap but low carrier mobility [14,15], which limits their device applications. However, phosphorene has a band gap of 1.5 eV [16,17] and high carrier mobility [18,19]. Since its experimental realization in 2014 [20,21], many studies have been conducted to explore the possible potential applications in electronic devices. For example, Landau levels have been considered [22,23,24]; the effects of strain on the optical properties [25] and thermoelectric properties [26] have been addressed elsewhere; and high-performance field-effect transistors based on phosphorene have been reported [27].

Nanoribbons of two-dimensional materials have many interesting electronic properties due to the more tunable edge states and quantum confinement effects. The electronic properties of phosphorene nanoribbons have attracted considerable attention [28,29,30]. One unique characteristic of phosphorene nanoribbons is the emergency of a quasi-flat band in the band gap of zigzag nanoribbons [31,32]. Ezawa [33] and Sisakht et al. [34] have explained the presence of the quasi-flat band in the tight-binding model; it was shown that the quasi-flat band can be controlled by a transverse electric field along the width of the nanoribbon. The individual two-dimensional phosphorene nanoribbons have been successfully synthesized by Watts et al. [35], which has already stimulated the experimental research on phosphorene-nanoribbon-based solar cells [36]. Graphene nanoribbons with zigzag edges are demonstrated to be unstable, but zigzag phosphorene nanoribbons are stable.

Spin-dependent thermoelectric transport is a new research field combining both spintronics and thermoelectricity [37], which has potential applications in low-power-consumption devices and environmental protection. The spin Seebeck effect [38], driven by a temperature gradient across the material, generates spin current without applying electrical gate voltages. The spin Seebeck effect has been observed in many experimental studies [39,40], and it is also theoretically predicted in graphene [41,42].

Due to its band gap, phosphorene has been regarded as a promising candidate for thermoelectric devices [43,44]. In their experimental study, Flores et al. [45] reported that the Seebeck coefficient for bulk black phosphorus can be as large as 335±10 μV/K. Ma et al. [46] obtained a high thermopower in zigzag phosphorene nanoribbons by applying gate voltages to the ribbon edge. We want to point out that the Seebeck coefficients for pristine graphene or silicene nanoribbons are much smaller than phosphorene nanoribbons. Many studies have been carried out to enhance the Seebeck coefficient in graphene or silicene; for example, the enhancement of Seebeck coefficients has been studied in strained graphene [47,48], graphene quantum rings [49], graphene with antidots [50,51], and deformed silicene [52]. The band gap and edge states offer phosphorene many advantages in thermoelectric applications, and it is much easier to enhance the Seebeck coefficient by modulating the edge states. On the other hand, phosphorene is also a promising material for spintronic-based devices because of its long spin lifetimes [53]. It is reported that the exchange splitting of 0.184 eV can be obtained in phosphorene by depositing a EuO ferromagnetic stripe on it [54].

In this study, we investigated the spin-dependent thermoelectric properties of zigzag phosphorene nanoribbons in the presence of a ferromagnetic stripe. The ferromagnetic stripe was deposited on a zigzag phosphorene nanoribbon, inducing exchange splitting in phosphorene through the proximity effect. We found that both charge and spin Seebeck coefficients could be greatly enhanced by modifications of the edge states of zigzag phosphorene nanoribbons. Two methods are proposed in this study to modulate the edge transport in zigzag phosphorene nanoribbon: one is applying different voltages on the two ribbon edges; the other is including notches on the two edges. The transport gap is enlarged by the modulation of the edge states, which enhances the charge and spin thermopower almost twofold.

## 2. Model and Methods

The device under consideration is shown in Figure 1a. A zigzag phosphorene nanoribbon (scattering region) was connected to two semi-infinite leads with different temperatures, T+ΔT/2 and T−ΔT/2, respectively, where *T* is the equilibrium temperature of the system and ΔT is the temperature difference between the left and right leads. A ferromagnetic stripe was deposited on the scattering region, which will induce exchange splitting in phosphorene through proximity effects. Two gate voltages were applied to the two zigzag atomic chain on the ribbon edge, which is marked as a yellow region in Figure 1b. The side view of the crystal structure of phosphorene is shown in Figure 1c. In our numerical calculations, the dimensions of the device were chosen to be P = H = 40.

In the tight-binding approximation, the Hamiltonian function for phosphorene nanoribbons in the presence of a gate potential and an exchange field can be written as
(1)$$H=\sum_{i,j,\alpha }t_{ij}c_{i\alpha }^{†}c_{j\alpha }+M\sum_{i,\alpha }c_{i\alpha }^{†}\sigma_{z}c_{i\alpha }+V_{g}\sum_{i,\alpha }c_{i\alpha }^{†}c_{i\alpha }$$
where the summations run over all the lattice sites with $t_{ij}$ as the hopping energies between two sites, and $c_{i\alpha }^{†}$ creates an electron with spin α on site *i*. $V_{g}$ and *M* are the gate potential and exchange splitting, respectively, and $\sigma_{z}$ is the *z* component Pauli matrix. In Figure 1b,c, we show the top and side views of the monolayer phosphorene lattice structure, respectively. The hopping energies between a site *i* and its neighbors are shown in Figure 1b, and the suggested values of these hopping integrals are specified as *t_1_* = −1.220 eV, *t_2_* = 3.665 eV, *t_3_* = −0.205 eV, *t_4_* = −0.105 eV, and *t_5_* = −0.055 eV [55]. The exchange splitting was chosen to be 0.184 eV, in accordance with a previous study [54], and we also considered other values to check the effects of exchange splitting strength. Two gate potentials, $V_{g1}$ and $V_{g2}$, were applied on the atomic chains of the left and right ribbon edges, respectively, which is indicated as a yellow region in Figure 1b.

The Seebeck coefficient, also referred to as thermopower, is a measure of the magnitude of the thermoelectric voltage induced by the temperature difference between the two ends of the device. It is one of the most important properties of thermoelectric materials. The spin-dependent Seebeck coefficient $S_{\alpha }$ (thermopower) can be calculated in terms of the transmission coefficient
(2)$$S_{\alpha }=-\frac{k_{B}}{e}\frac{\int dE\left(-\frac{\partial f_{0}}{\partial E}\right)\frac{E-E_{F}}{k_{B}T}T_{\alpha }\left(E\right)}{\int dE\left(-\frac{\partial f_{0}}{\partial E}\right)T_{\alpha }\left(E\right)}$$
where $f_{0}=1/[e^{(E-E_{F})/k_{B}T}+1]$ is the Fermi distribution function and $T_{\alpha }(E)$ is the transmission coefficient. Using the nonequilibrium Green’s function (NEGF), the transmission coefficient from terminal *L* to terminal *R* is written as
(3)$$T_{\alpha }(E)=Tr\left\{\Gamma_{R}(E)G^{r}(E)\Gamma_{L}(E)G^{a}(E)\right\}$$
where $\Gamma_{R(L)}$ is the level broadening matrices written in terms of the self-energies of the leads $\Gamma_{R(L)}=i[\Sigma_{R(L)}^{r}-\Sigma_{R(L)}^{a}]$, and $G^{r(a)}(E)$ is the retarded (advanced) Green’s function. The Green’s function is calculated from the Hamiltonian of the scattering region and the interaction of the leads, which is given by
(4)$$G^{r(a)}={\left[EI-H-\Sigma_{L}^{r(a)}-\Sigma_{R}^{r(a)}\right]}^{-1}$$
where *H* is the tight-binding Hamiltonian of the scattering region and $\Sigma_{L}^{r(a)}\left(\Sigma_{R}^{r(a)}\right)$ is the self-energy due to the left (right) lead. All numerical calculations were performed using the Python package Kwant [56]. According to this definition, the energy-dependent transmission coefficient plays an important role; thus, the transport properties of electrons in the quasi-flat band can have a considerable influence on the Seebeck coefficient. The charge and spin Seebeck coefficients can be defined as
(5)$$\begin{array}{l} S_{ch}=(S_{↑}+S_{↓})/2\\ S_{sp}=(S_{↑}-S_{↓})/2 \end{array}$$
where $S_{↑}(S_{↓})$ is the Seebeck coefficient for spin-up (down) electrons, which was obtained from Equation (2). In the present study, we also considered the band structures and local density of states of the proposed device, which were also obtained from Kwant.

## 3. Results and Discussion

The band structures, transmission, and LDOS of the device are presented in Figure 2. In Figure 2a,b, we show the band structures for pristine and ferromagnetic zigzag phosphorene nanoribbon, respectively. We can see from Figure 2a that there is a quasi-flat-edge band in the energy region −0.311 eV<E<0, although it is composed of two separate bands when we focus on this quasi-flat-edge band, which can be seen in the inset of Figure 2a. In Figure 2b, we can see that the exchange splitting of the ferromagnetic stripe lifts the degeneracy of the edge bands, and the spin-up bands are shifted upward (red), whereas the spin-down bands are shifted downward (blue). In Figure 2c, we show the transmission of the device. In the energy region −0.311 eV<E<0, the transmission is contributed from the quasi-flat-edge band, and the transmission for spin-up and spin-down electrons are found at different energies. We can see that the transmission oscillates with the energy, which is caused by the electron scatterings in the scattering region. When electrons move into the scattering region, electrons are backscattered in the scattering region, which produces the transmission oscillations.

In Figure 2d−f, we plot the LDOS of the device at energy E=−0.25  eV (the dashed green line). In Figure 2d, we show the total LDOS, including spin-up and spin-down electrons, and it is found that the LDOS is localized near the ribbon edges. This means that electrons from the quasi-flat band only move on the edges, which offers us a possibility to control the transport by modifying the edge states. In Figure 2e,f, we show the LDOS for down spin and up spin, respectively. We can see that electrons move at the edge, and only spin-down electrons can transport from the left lead to the right lead.

In the following discussions, we consider the spin-dependent Seebeck coefficients for a phosphorene nanoribbon with a ferromagnetic stripe; then, we show the enhancement of Seebeck coefficients via the modifications of the edge states of zigzag phosphorene nanoribbons. In Figure 3a, the spin-up and spin-down Seebeck coefficients are plotted as a function of the Fermi energy. It is noted that both spin-up and spin-down Seebeck coefficients oscillate with the Fermi energy with different dependences. The amplitudes of Seebeck coefficients are large in the energy region $-1.2\;\mathrm{eV}<E_{F}<-0.3\;\mathrm{eV}$. The positive peak height is around $14\;k_{B}/e$, and the negative peak height is around $-13.1\;k_{B}/e$. The spin and charge Seebeck coefficients are shown in Figure 3b. We can see that the amplitudes of the spin Seebeck coefficient can be as large as $13.2k_{B}/e$, and the amplitude of the charge Seebeck coefficient can reach $11.6k_{B}/e$. Moreover, we could obtain some energy points that had spin a Seebeck coefficient but zero charge Seebeck coefficient. For example, the spin Seebeck coefficients for points P and Q are $13.2k_{B}/e$ and $6.4k_{B}/e$, respectively, but the corresponding charge Seebeck coefficients are zero.

It has been reported in some previous investigations [57] that the enlargement of the transport band gap can enhance the Seebeck coefficients. The quasi-flat-edge band lies in the band gap; thus, so the modulations of the edge states of the zigzag phosphorene ribbon may enhance the Seebeck coefficient in our device. Here, we modulated the band gap by applying two gate voltages on the left and right edges of the zigzag ribbon (yellow region in Figure 1b. Firstly, we only applied one gate voltage on the left side of the ribbon, and the corresponding band structure and spin-dependent coefficients are shown in Figure 4(a1–a3). It is noted in Figure 4(a1) that the gate voltage lifts the degeneracy of the edge modes, and both the spin-up and spin-down edge modes on the left side of the ribbon are shifted upward. The edge state bands on this ribbon edge merge into the bulk conduction band. Comparing the Seebeck coefficients shown in Figure 4(a2,a3) with Figure 3, we can see that the peak heights of the Seebeck coefficients only have very small changes, because the band gap is not actually opened although two edge state bands are moved out of the gap. In Figure 4(b1–b3), we show the band structure and Seebeck coefficients for the device with two different gate voltages applied on the left and right edges. The transmission in the energy region −0.311 eV<E<0 (see Figure 2c) becomes zero, which means that the transport gap is enlarged. We can see from Figure 4(b2) that the spin-dependent Seebeck coefficient can be as large as $22.6\;k_{B}/e$, which is a large enhancement of the thermopower. In Figure 4(b3), it is noted that the peak heights of the spin and charge Seebeck coefficients are $22.3\;k_{B}/e$ and $22.6\;k_{B}/e$, respectively. We can find a point with very high spin Seebeck coefficient but zero charge Seebeck coefficient. The spin Seebeck coefficient is $22.2\;k_{B}/e$ at point P, but the charge Seebeck coefficient is zero. It is obvious that the spin-dependent Seebeck coefficients can be considerably enhanced when the edge states are modified by the gate voltages. We also considered the spin-dependent Seebeck coefficient of the device with ferromagnetic stripes deposited on the scattering region and the leads; it was found that the Seebeck coefficients can still be enhanced via modifications of the edge states in zigzag ribbons. We only observed small changes in the magnitude of the spin and charge Seebeck coefficients.

As already mentioned in Section 2, the Seebeck coefficient is determined by the transmission coefficient, and only the electrons within several $k_{B}T$ around the Fermi level contribute to the Seebeck coefficient. In a previous study [58], the authors found that the conductance plateau contributed from the quasi-flat-edge band can be decreased by notching atoms from the zigzag edges of the ribbon. Here, we plotted the Seebeck coefficients as a function of Fermi energy when notches were considered on the ribbon edges. Firstly, we considered a square notch on one edge of the ribbon (see Figure 5(a1)). The peak heights of the Seebeck coefficients only exhibited a very small change compared with the results shown in Figure 3. For example, the peak heights in Figure 5(a2,a3) are $14.2\;k_{B}/e$ and $13.1\;k_{B}/e$, which are almost the same as that in Figure 3. In Figure 5(b1−b3), we show the spin-dependent Seebeck coefficients for the ribbon with notches on both edges. The constrictions in the scattering region affect the scatterings and suppress the transmission contributed from the edge states. The transmission in the energy region −0.311 eV<E<0 (see Figure 2c) becomes zero due to the notches on the edges. It is noted in Figure 5(b2) that the peak heights of spin-dependent coefficients are largely enhanced. For example, the positive peak heights for spin-up and spin-down coefficients are $19.2\;k_{B}/e$ and $23.3\;k_{B}/e$, respectively, which are largely enhanced compared with the results shown in Figure 3a and Figure 5(a2). We can also see that the spin and charge coefficients shown in Figure 5(b3) are largely enhanced by modulating the edge transports on both edges of the ribbon.

The exchange splitting and temperature have been chosen to be *M* = 0.184 eV and $k_{B}T=0.03\left|t_{1}\right|$, respectively. Here, we consider the effects of exchange splitting strength and temperatures. In Figure 6(a1−a3), we justify the value of exchange appropriately, and three values of exchange splitting, *M* = 0.05 eV, 0.1 eV, and 0.25 eV, are considered. This is useful for future experimental studies, because other stronger or weaker ferromagnets may be chosen in the experiments. We can see that very small changes are found in charge Seebeck coefficients when we change the exchange splitting strength. The peak heights of the charge Seebeck coefficient are around $22.6k_{B}/e$ for all the exchange splitting strengths. However, it is noted that the spin Seebeck coefficients are changed greatly due to the change in exchange splitting strength. We can see in Figure 6(a1−a3) that the spin Seebeck coefficient is very small when the exchange splitting is very small. When the exchange splitting M>0.1 eV, the spin Seebeck coefficient is larger than $13k_{B}/e$. This means that we do not need very strong ferromagnets in the experiments, and this theoretical prediction can be realized in future experimental studies. The temperature $k_{B}T=0.03\left|t_{1}\right|$ can also be expressed as *T* = 424 K, and we extend it to *T* = 200 K, 300 K, 500 K. We can see that the spin-dependent Seebeck coefficient is larger when we have a lower temperature; for example, the peak heights for *T* = 300 K is around (see Figure 6(b2)), which is larger than that of Figure 4(b3). For a higher temperature *T* = 500 K, the peak heights of Seebeck coefficients decrease slightly. It is no doubt that the device can be realized at room temperature.

## 4. Conclusions

In conclusion, we have investigated the spin-dependent thermoelectric transport in zigzag phosphorene nanoribbons with a ferromagnetic stripe. We reveal that phosphorene nanoribbons could be very promising for spin caloritronics applications. We calculated the spin-dependent Seebeck coefficients of phosphorene nanoribbons in the presence of exchange splitting fields induced by the proximity of ferromagnetic materials. The main findings are that the spin-dependent Seebeck coefficient can be enhanced by modulating the edge states of the zigzag phosphorene nanoribbon. We proposed two methods to modulate the edge-state transport to enhance the Seebeck coefficients: one method is adding different gate voltages on the edges; the other is including notches on the ribbon edges. It was found that the transport gap opened between the valence and conduction bands, and the Seebeck coefficients were enhanced almost twofold. The findings in this study will be useful for the design of tunable spin-dependent thermoelectric devices, and it will stimulate more interesting investigations of spin-thermoelectric phenomena in phosphorene nanoribbons.

## Figures and Tables

**Figure 1 nanomaterials-12-02350-f001:**
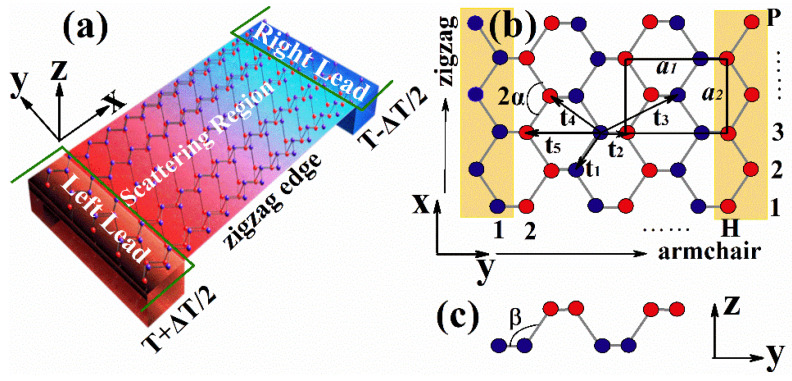
Schematic illustration of the proposed thermoelectric device. (**a**) A zigzag phosphorene nanoribbon is attached to hot and cold macroscopic reservoirs. (**b**) Top and (**c**) side views of the crystal structures of monolayer phosphorene. The hopping integrals, $t_{i}$, between the phosphorus atoms are shown. The dimensions of a ribbon can be determined by number of atoms on the edges H and P.

**Figure 2 nanomaterials-12-02350-f002:**
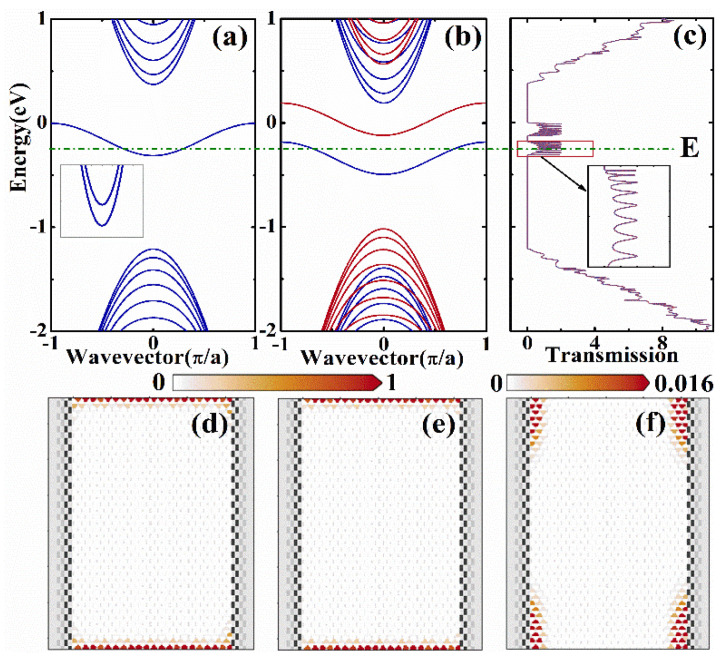
The band structures for (**a**) pristine zigzag phosphorene nanoribbon and (**b**) ferromagnetic zigzag phosphorene nanoribbon. (**c**) The transmission of the device. (**d**–**f**) LDOS of the device for total, spin-down and spin-up electrons, respectively, with E = −0.25 (green dashed dotted line).

**Figure 3 nanomaterials-12-02350-f003:**
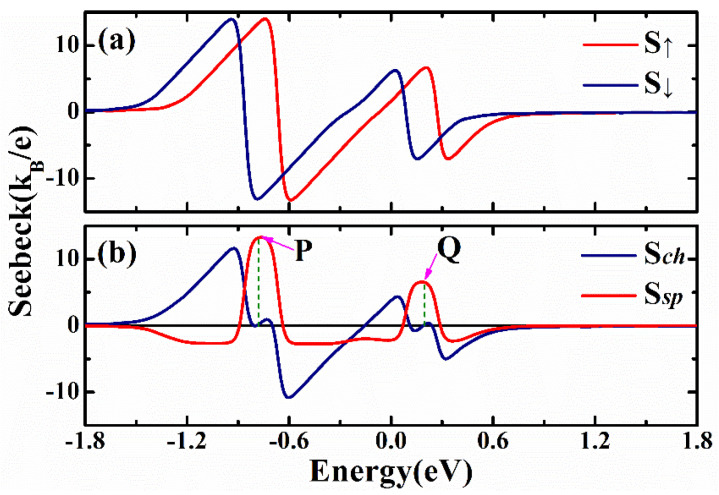
(**a**) Spin-dependent Seebeck coefficient for a zigzag phosphorene nanoribbon with a ferromagnetic stripe. (**b**) The corresponding charge and spin Seebeck coefficients. The parameters for the device are: gate voltages $V_{g1}=V_{g2}=0$, exchange splitting M=0.184 eV, and temperature $k_{B}T=0.03\left|t_{1}\right|$.

**Figure 4 nanomaterials-12-02350-f004:**
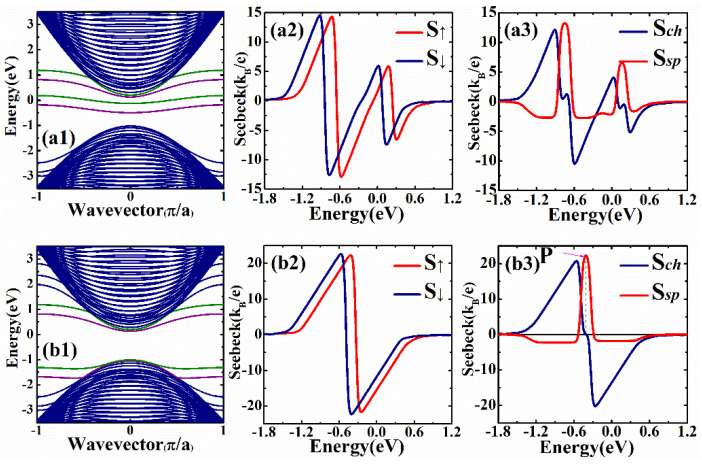
The band structures and spin-dependent Seebeck coefficients for the device with different voltages applied on the left and right edges of the ribbon. (**a1**–**a3**) The gate voltage is only applied on left edge with $V_{g1}=1\;\mathrm{eV}$ and $V_{g2}=0\;\mathrm{eV}$. (**b1**–**b3**) The gate voltages are applied on both edges with $V_{g1}=1\;\mathrm{eV}$ and $V_{g2}=-1.5\;\mathrm{eV}$. The other parameters for the device are: exchange splitting M=0.184 eV and temperature $k_{B}T=0.03\left|t_{1}\right|$.

**Figure 5 nanomaterials-12-02350-f005:**
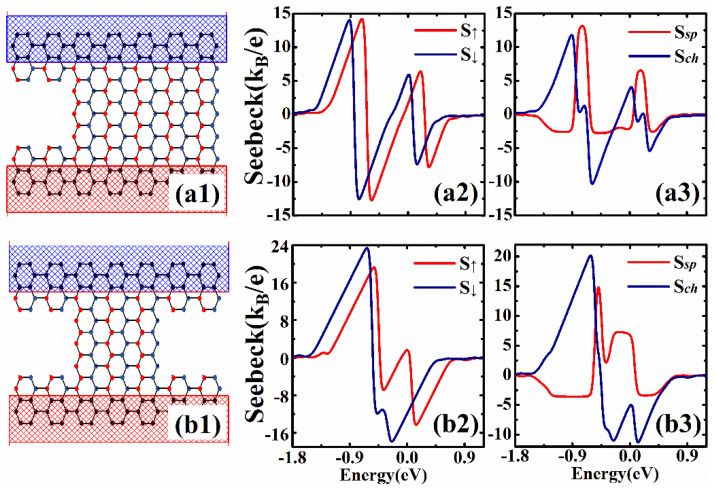
Spin-dependent Seebeck coefficients for the device with notches on the edges of the ribbon. (**a1**–**a3**) The schematic illustration and spin-dependent Seebeck coefficients for the device with one asymmetric square notch on the ribbon edge. (**b1**–**b3**) The schematic illustration and spin-dependent Seebeck coefficients for the device with two square notches on the two ribbon edges. The other parameters for the device are: exchange splitting M=0.184 eV and temperature $k_{B}T=0.03\left|t_{1}\right|$.

**Figure 6 nanomaterials-12-02350-f006:**
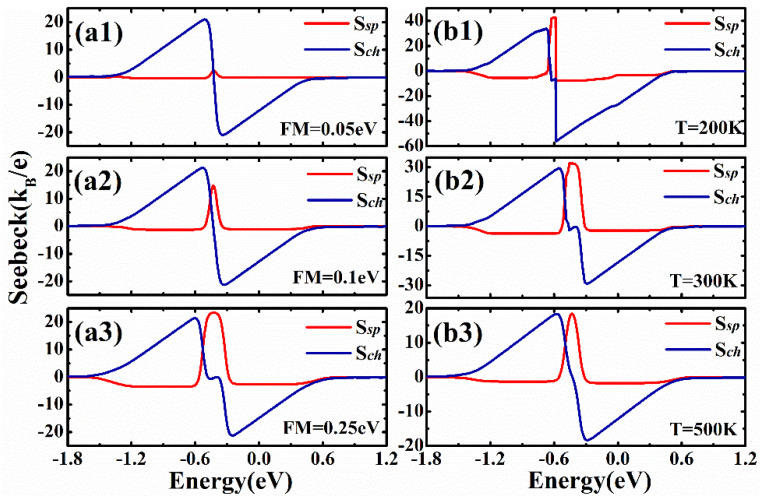
The effects of exchange splitting strength and temperature on the spin-dependent Seebeck coefficients. The gate voltages were applied on both edges and all the parameters are the same as in Figure 4. (**a1**–**a3**) The spin and charge Seebeck coefficients for $k_{B}T=0.03\left|t_{1}\right|$ and M=0.05 eV, 0.1 eV, 0.25 eV. (**b1**–**b3**) The spin and charge Seebeck coefficients for *M* = 0.184 eV and T=200 K, 300 K, 500 K.

## Data Availability

The data presented in this study are available upon reasonable request from the corresponding author.

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
