# Peer review of "Enhanced Spin Thermopower in Phosphorene Nanoribbons via Edge-State Modifications"

_nanomaterials, 2022, doi:10.3390/nano12142350_

Round 1
Reviewer 1 Report
This manuscript by J. Ou and Q. Zhang reports a computational study of the spin-dependent thermoelectric transport properties of zigzag phosphorene nanoribbons placed in the vicinity of a ferromagnetic stripe. On one hand, the effect of applying gate voltages at one or both edges is explored. On the other hand, with the aim of enhacing the thernoelectric power of the ribbons, two different strategies based on the modification of the edge states are proposed.
Although the systems and topic under consideration are interesting, the system and method considered for the simulations are very poorly described. Moreover, the quality of the simulations is questionable and the analysis of the results unsatisfactory. For all there reasons, I believe the manuscript does not meet the criteria for its publication in Nanomaterials.
To further sustain my decision, and in the hope that it might be helpful to the authors, some specific comments are given in the following:
1) In section 2, “Model and methods”, the right/left leads and the scattering region should be clearly illustrated. The choice of a givenvalue (0.184eV) for the exchange splitting M should be justified appropriately. The method, and not only the code, used for calculating the electronic transmission should be described. The authors claim that a ferromagnetic stripe is deposited on the device region but no further details are given: does it mean that this effect is only included in the scattering region but not in the leads? If so, how is the matching between the scattering region and the leads done? In particular, in the case of semiconducting electrodes, the level alignment between scattering region and leads is very difficult to achieve. In relation to Eq. (2) they say that “the Seebeck coefficient is determined by the transmission-weighted-average value of energy”; this description is very confusing.
2) Regarding the results in Figure 2, these correspond to periodic systems. Thus, I suppose it represents the electronic band structure of the leads. However, in the caption it says that panel (b) shows the results for the nanoribbons with an exchange field, which in section (1) authors claim is only deposited in the device region. Panels (d)-(e) are unclear both from the image point of view as well as from the explanation of their content.
3) A clear connection between the modification of the edge states and the way in which this affects the Seebeck coefficient is not discussed. Rather than just giving an account of results, a thorough analysis of the results would be desirable. For instance, the modifications considered in Figure 5 give rise to nanoribbon constrictions which still possess a zigzag edge conformation. However, the authors claim that those indentations destroy the edge state.
Author Response
Dear Reviewer,
We greatly appreciate your thoughtful comments and constructive suggestions that helped improve the manuscript. We have addressed all the questions mentioned in the comments and revised the manuscript carefully, in response to all the comments and suggestions. Please find our responses to the comments and the list of changes in the manuscript. We are looking forward to hearing from you again.
Yours sincerely,
Qingtian Zhang

Reviewer 2 Report
In this work, the authors theoretically investigated the thermoelectric transport and its dependence on spin in phosphorene ribbons with edge modification. The authors presented calculated results of the band structure and Seebeck coefficients for two different types of edge defects and concluded that the edge-state modifications can enhance the spin thermopower. I have a few comments as follows.
(1) The Seebeck effect is well known to be dependent on temperature. But here the authors ignore the temperature-dependence in the current study. Is there any specific reason for it?
(2) More discussion on the physical picture behind the calculated data is necessary. For example, what is the meaning of spin up/down separation? What is its significance for experimental measurements as well as technological applications? What is the connection between the Seebeck coefficient and transmission mechanism in material? Why is it important to understand the Seebeck coefficient?
(3) A comparison between the presented results for P ribbons and other 2D materials such as graphene, silicene in the literature can help enhance the motivation of this study.
(4) Zigzag-edge ribbons have been demonstrated to be unstable structures. The author might reconsider their results and significance of this work.
Author Response
Dear Reviewer,
We greatly appreciate your thoughtful comments and constructive suggestions that helped improve the manuscript. We have addressed all the questions mentioned in the comments and revised the manuscript carefully, in response to all the comments and suggestions. Please find below our responses to the comments and the list of changes in the manuscript. We are looking forward to hearing from you again.
Yours sincerely,
Qingtian Zhang

Round 2
Author Response
Dear Reviewer,
We greatly appreciate your constructive suggestions and efforts towards improving our manuscript. We have addressed all the questions and revised the manuscript carefully, in response to all the comments and suggestions. Please find below our responses to the comments and the list of changes in the manuscript. We are looking forward to hearing from you again.
Yours sincerely,
Qingtian Zhang

Reviewer 2 Report
Now the manuscript is improved a lot and serves the important subject
of thermoelectric transport. I recommend the present manuscript to be
published in Nanomaterials.
Author Response
Dear Reviewer,
We greatly appreciate your constructive suggestions and efforts towards improving our manuscript. Thank you very much.
Yours sincerely,
Qingtian Zhang
Round 3
Reviewer 1 Report
I acknowledge the authors' effort in clarifying the issues I pointed out, and I consider the manuscript now meets the criteria for its publication in Nanomaterials.